# Immunonutrition in Operated-on Gastric Cancer Patients: An Update

**DOI:** 10.3390/biomedicines12122876

**Published:** 2024-12-18

**Authors:** John K. Triantafillidis, Konstantinos Malgarinos

**Affiliations:** 1“Metropolitan General” Hospital, Holargos, 15562 Athens, Greece; 2Hellenic Society for Gastrointestinal Oncology, Iera Odos 354, Haidari, 12461 Athens, Greece; konstantinosmalgarinos@gmail.com

**Keywords:** gastric cancer, immunonutrition, nutrition, perioperative nutrition

## Abstract

Enteral immune nutrition has attracted considerable attention over the past few years regarding its perioperative role in patients undergoing major surgery for digestive cancer. Today, the term enteral immune nutrition refers to the perioperative administration of nutritional preparations containing, among others, specific ingredients such as glutamine, omega-3 polyunsaturated fatty acids, and arginine. They provide nutritional support and exert pharmacological effects through the substances contained in these preparations. Their administration to patients with gastric cancer is necessary as malnutrition and other metabolic disorders are frequent symptoms with effects on the level of immune responses, affecting the function of intestinal permeability and, therefore, the effectiveness of chemotherapy. Existing clinical data and data from all meta-analyses published so far support the view that enteral immune nutrition enhances the immune responses of gastric cancer patients, and reduces the rate of postoperative complications, and the duration of hospitalization without, however, improving patient survival. The content of enteral immune nutrition, dose, administration interval, and the effect on patient survival should be more precisely determined through relevant extensive multicenter studies. This systematic review describes and analyses the clinical results and the findings of relevant meta-analyses of the application of enteral immune nutrition in gastric cancer patients, emphasize the importance of this therapeutic intervention for disease progression, and attempts to provide practical guidelines for applying enteral immune nutrition in daily clinical practice.

## 1. Introduction

Enteral Immune Nutrition (EIN) is the ability to regulate and modify the human immune system’s function through interventions carried out by the administration of specific nutrients. The ESPEN guidelines on nutrition in cancer patients refer to arginine, omega-3 fatty acids, and nucleotides under the heading “immunonutrition” [1]. According to this definition, EIN can be applied to a variety of conditions in which the modification of immune responses is required to achieve therapeutic benefit. It is understood that the term EIN encompasses issues related to a variety of parameters such as immunity, infections, inflammation, and tissue damage [2].

The individual elements that make up the EIN are the amino acids glutamine and arginine, polyunsaturated (omega-3) fatty acids, nucleotides, taurine, beta-carotene, trace elements such as zinc and selenium, anti-oxidants, copper, and vitamins A, B, C and E. Therefore, the EIN is a solution high in protein and energy, enriched with specific nutrients [3]. EIN preparations are divided into those enriched in amino acids, such as glutamine and arginine, and those with a high content of omega-3 fatty acids (ω-3PUFAs) intended for surgical patients. There are also formulas with higher doses of ω-3PUFAs omega-3 fatty acids and reduced amino acid content intended for cancer patients. At present, the solution is administered orally or via a gastric catheter because there are no EIN solutions for parenteral use. 

EIN is applied in various clinical situations, and it is expected to reduce the risk of postoperative complications and hospitalization. It is well established that surgical procedures performed on cancer patients for the treatment of the underlined malignant disease, result in reduced immune responses with an increase in mortality and morbidity associated with the occurrence of postoperative complications, and especially postoperative infections. Therefore, if there was a way to restore immune competence, at least partially, one would expect a reduction in the incidence of these complications. This is achieved to some extent with EIN through artificial nutrition products and nutrient-enriched solutions, which also aim to restore the postoperative nitrogen balance and increase protein synthesis.

The main constituents of EIN are the following:

The amino-acid Glutamine has been shown to be vital for maintaining immune system function, nitrogen balance, intestinal integrity, and fighting infections [4]. In an experimental GC model, oral glutamine administration decreased tissue glutathione levels, increased plasma glutathione levels, enhanced T lymphocyte proliferation, increased NK cell activity, suppressed TNF-α secretion, and increased IL-2 secretion [5]. Glutamine also plays a role in intracellular signaling by increasing the expression of heat shock proteins, inhibiting apoptosis, and reducing inflammatory responses [6]. During catabolic states, glutamine depletion increases the risk of secondary infections, prolongs recovery time, and increases mortality rates. Supplementation of glutamine in critically ill or surgical patients has been shown to reduce the incidence of hospital-acquired infections, complications, and hospitalization time [7]. There is considerable variation in the ratio and concentration of glutamine in EIN formulations. A randomized controlled trial did not demonstrate significant differences in postoperative nutrition, hepatic and renal function, or recovery between alanine-glutamine ratios. However, the higher alanine-glutamine ratio (30% of the total amino acids in parenteral nutrition) demonstrated better effects on T lymphocyte modulation [8]. 

Arginine, a non-essential amino acid (becoming however conditionally essential during critical illness), is considered to be a precursor of polyamides and nucleosides, stimulating growth hormone production. It has anabolic activity and increases the number of T-helper cells [9]. Arginine-containing dietary supplements can increase protein and collagen deposition in experimental wounds [10]. Arginine may also regulate postoperative inflammatory responses and immune function by increasing the concentration of immunoglobulin IgM [11]. Finally, arginine plays a role in activating anticancer immunity and regulating the efficacy of immunotherapy, mainly through two metabolic pathways involving arginase and nitric oxide. Nitric oxide, a metabolic product of the body is a free radical that helps maintain microcirculation and eliminate microorganisms. Nitric oxide also exhibits anti-tumor effects enhancing chemotherapy sensitivity by regulating tumor vascular function [12]. However, clinicians should be cautious when supplementing arginine, as excessive nitric oxide production caused by inducible nitric oxide synthase can lead to tissue damage. In cases of severe inflammation, arginine supplementation may exacerbate the inflammatory state. Therefore, the use of arginine should be carefully considered in clinical applications, with attention to optimal dosage, timing, and patient conditions [13]. It is, however, unexplained as to why lysine, an also basic amino acid, displays comparatively reduced immunological activity. It is well known that the two most important features through which cancerous tumors ensure their survival are the ability to evade the host immune system and metabolic reprogramming. Lactic acid and acidification of the tumor microenvironment favor the processes of carcinogenesis. In particular, histone lysine lactylation is a novel post-modification that gradually reveals its crucial role in tumor biology, the role of lactate as an immunosuppressive molecule, and the effects of histone lysine or non-histone acetylation on immune cells. Lysine is, therefore, a critical molecule that enhances the malignant potential of tumors by affecting the tumor microenvironment and orchestrating the transitions of immune states [14,15]. 

Ω-3 PUFAs are essential fatty acids rich in alpha-linolenic acid, docosahexaenoic acid, and eicosapentaenoic acid found mainly in fish oil. A number of mechanisms are involved in the anti-inflammatory action of ω-3PUFAs omega-3 fatty acids, the most important being the reduction in eicosanoid synthesis, the reduction in oxidative stress, and the expression of inflammatory cytokines, such as IL-6 and IL-10, thus helping to resolve inflammation [16]. This assumption is further supported by the findings of a trial showing that fish oil supplementation had a positive effect on inflammatory markers in GC patients undergoing surgery [17]. Another study showed that postoperative application of ω-3PUFAs omega-3 PUFAs could enhance NK cell activity, helping to maintain the immune response at a moderate level of sensitivity through methylation of the TNF-α gene promoter [18]. Ω-3PUFAs Omega-3 PUFAs also protect the intestinal epithelial barrier and the integrity of the gut microbiota composition, reducing bacterial translocation and preventing postoperative complications such as intestinal leaky gut syndrome [19]. A randomized controlled trial observed that a diet rich in omega-3 PUFAs significantly increased the abundance of several genera producing short-chain fatty acids, including *Bifidobacterium*, *Roseobacterium*, and *Lactobacillus* [20]. This favorable effect is supported by the results of a relevant meta-analysis [21]. It is interesting to note that the simultaneous use of prebiotics, ω-3PUFAs n-3 PUFAs, and EIN increases the structural and functional integrity of the intestinal mucosal barrier [22]. The role of other constituents of EIN such as nucleotides and selenium, which represent vital molecules for the regulation of glutathione peroxidase, is less clear.

Several studies and meta-analyses have been published in recent years on the role of perioperative IEN administration in several malignant neoplasms, particularly in digestive tract neoplasms. Especially for gastric cancer (GC) the results are quite encouraging in terms of postoperative complications and hospital stay, however, without any impact on patient survival. Therefore, the aim of this study was to describe the existing experimental and clinical data concerning the therapeutic efficacy of perioperative administration of EIN in patients with GC, to emphasize the importance of this therapeutic intervention for disease progression, and to provide practical guidelines regarding the administration of EIN in daily clinical practice. 

## 2. Methods

The study was conducted according to PRISMA guidelines and registered in the Open Science Framework (OSF, registration DOI https://doi.org/10.17605/OSF.IO/G8TYS). Pubmed, EMbase, and The Cochrane Library databases were searched up to 2024. Search terms included were “immunonutrition”, “enteral immune nutrition”, “gastric cancer”, and “stomach cancer”. Only articles published in English were evaluated. All clinical studies related to the effects of EIN on the immune responses of GC patients, as well as all clinical trials and meta-analyses related to the effect of EIN on the clinical course of patients with GC, were included and analyzed. Therefore, the identified studies were categorized into three parts according to their topic and task: (i) clinical studies, i.e., studies investigating the role of EIN in GC patients; (ii) studies comparing EIN with standard enteral nutrition (SEN), and (iii) meta-analyses regarding the efficacy of EIN in GC surgical patients. The relevant literature was reviewed, and data from each study were recorded. All prospective and retrospective clinical studies were selected. In all studies, the following data were recorded: author, year of publication, number of patients, characteristics of the patient population, type of diet applied, results, and conclusions. The number of records identified from relevant databases was 668. A total number of 586 articles were excluded because they were irrelevant to the subject of the study. After exclusion of these studies, a total number of 82 articles remained for evaluation of which 74 were assessed for eligibility (8 studies were excluded due to unavailable details). From this number 44 were finally excluded for various reasons (reviews: 13, other languages: 13, and other reasons: 18), thus leaving 30 studies for final analysis (clinical studies: 10, systematic reviews and meta-analyses: 11, and studies dealing with the effects of EIN in immune responses of GC patients: 9 studies). Flow information through the different phases of this systematic review is given in Figure 1.

## 3. Results

The results will be presented in three subsections: (i) studies investigating the effect of ΕΙN on patients’ immune responses, (ii) results of clinical studies, and (iii) results of meta-analyses.

### 3.1. Effect of IEN on Patients’ Immune Responses

Regarding the effects of EIN on the immune responses and the level of immune defense of GC patients, the existing data are as follows: Molfino A et al. investigated the effect of EIN on inflammatory infiltration of the tumor microenvironment using immunohistochemical analysis in 12 patients undergoing surgery for GC, both preoperatively and postoperatively. They used immunohistochemical analysis in the tumor microenvironment in four patients, as well as: in five patients receiving nutritional supplementation and in three patients without any kind of nutritional support. They found differences between the three groups, however, none reached significant levels. More specifically, an increase in the number of CD8+ T-lymphocytes and an increase in the number of CD83+ and CD68+ was found in the EIN group. Also, in the EIN group, a positive correlation between CD8+ and CD68+ macrophages and a significant correlation between CD68+ and CD40+ was observed. Therefore, administration of EIN to GC patients probably induces changes in the tumor microenvironment [23]. In their study, Ma M et al. evaluated the effect of EIN on postoperative immune competence and intestinal mucosal barrier function in 30 GC patients undergoing radical gastrectomy, in comparison with 35 control patients who did not receive EIN. On postoperative day three and up to postoperative day seven, the levels of transaminoxidase, D-lactic acid, and endotoxin in the EIN group were significantly lower than those in the control group. The time periods for first gas expulsion and postoperative complications were significantly shorter in the EIN group. Furthermore, on postoperative day seven, white blood cell count, and neutrophil/lymphocyte ratio were lower in the EIN group, while albumin levels were higher. Interestingly, one month after surgery, CD4+T and CD8+T counts were significantly higher in the EIN group. This study clearly showed that perioperative administration of EIN enhances immune responses and intestinal barrier in GC patients undergoing radical gastrectomy [24]. Xu LN et al. investigated the effect of EIN with ω-3PUFAs omega-3 fatty acids on the methylation and function of natural killer (NK) cell genes in 70 elderly GC patients separated into the EIN with ω-3PUFAs omega-3 fatty acids group and the placebo group. After 14 days of EIN, patients in the ω-3PUFAs omega-3 fatty acid group had significantly higher mean NK cell activity and lower TNF-α gene promoter methylation rates than the placebo group. No significant differences were observed in serum albumin, proalbumin, TNF-α levels, and NK cell counts between the two groups. Therefore, postoperative application of EIN with ω-3PUFAs omega-3 fatty acids improves NK cell activity, which correlates with the methylation status of the TNF-α gene promoter [18]. Franceschilli M et al., in a retrospective study, investigated the effects of EIN (immunosupplementation with maltodextrin on the day of surgery) compared to patients who underwent the same surgery but without EIN. After one month, postoperative complications were 8.7% in group A and 33.3% in group B, respectively, and the mean hospital stay was significantly shorter in the EIN group [25]. In a prospective study in surgical patients with GC, Dias et al. the authors evaluated the effect of EIN on patients’ nutritional status, levels of inflammatory markers, and changes in immune markers. They found a preoperative increase in CRP and IL-6 and a postoperative increase in the CD4:CD8 ratio. Complications and death were observed at 35%, mainly in patients with high preoperative IL-6, lower CD4:CD8 ratio, and lower protein and calorie intake. The authors conclude that high-calorie and protein supplementation combined with EIN preserves GC patients’ nutritional and immunological status [26]. One of the most common complications of GC surgery patients, especially those with severe nutritional deficiencies, is delayed healing or even non-healing of the surgical wound. In order to further clarify this issue, Farreras N et al. investigated the effect of early postoperative IEN on the rate of surgical wound healing in sixty six GC patients. Patients were randomized to an early postoperative EIN group containing arginine, ω-3PUFAs omega-3 fatty acids, and ribonucleic acid (30 patients) or a control group who received an isocaloric-isonitrogenous diet (36 patients). The wound healing process was evaluated by quantifying hydroxyproline deposition in a subcutaneously placed catheter and recording the complications of surgical wound healing. Patients in the EIN group had higher local hydroxyproline levels and significantly fewer surgical wound-healing complications episodes than those in the control group [27]. An earlier study also found that perioperative administration of an immune-enhanced diet significantly reduces systemic perioperative inflammation and postoperative complications in patients undergoing major surgery for intra-abdominal cancer compared to administration of a postoperative diet alone [28]. Okamoto Y et al., in agreement with all published studies, found that the incidence of postoperative infectious complications in the group of GC patients receiving an immune-enhancing diet was significantly lower compared to the conventional diet group. Also, the duration of the systemic inflammatory response syndrome group receiving an immune-enhancing diet was considerably shorter than that of the traditional diet group. Finally, the postoperative CD4(+)T-cell count on preoperative day one and postoperative day seven was significantly higher in the immune-enhancing diet group than in the conventional diet group. This study demonstrates that preoperatively administered immune-enhancing formulations containing arginine and ω-3PUFAs omega-3 fatty acids improve patients’ immune status, reduce the duration of systemic inflammatory response syndrome, and reduce the incidence of postoperative infectious complications [29]. Finally, Kamocki ZK et al., found severe impairment of the thrombocyte phagocytic activity in GC patients, a reduction which was partially improved as a result of perioperative EIN in all patients irrespective of the stage of GC [30]. 

### 3.2. Clinical Trials 

Perioperative EIN has been studied in randomized controlled trials. Most of these trials found EIN to be beneficial and associated with a significant reduction in the infectious complication rate, length of hospital stay, and health cost compared to a standard isocaloric, isoenergy nutritional solution (SEN). It is worth noticing, however, that some of these studies suffer from heterogeneity in terms of the state of malnutrition of patients at inclusion, the variety of products used, the prescribing periods, and occasionally from small patient numbers. Nevertheless, a cardinal feature of almost all the studies, with few exceptions, is that EIN is superior to SEN in many parameters, although no study has detected increased patient survival. It is noteworthy, however, that no study assessed the patient’s quality of life, which could significantly strengthen the arguments of those who support the perioperative administration of EIN in these patients.

Table 1 shows the relevant studies, the results, and the conclusions we can draw. In more detail, Yu J et al. randomized 112 patients with GC and cachexia to receive preoperatively either EIN (n = 56) or SEN support (n = 56) in a 1:1 ratio. They found that the incidence of postoperative infectious complications and the rate of overall complications were significantly lower in the EIN group compared to the SEN group. Furthermore, patients in the EIN group had significantly lower white blood cell, CRP, and IL-6 levels and higher lymphocyte counts and immunoglobulin IgA levels than those in the SEN group. Furthermore, patients in the EIN group received antibiotics for a shorter period, had shorter hospital stays, lost less weight, and had lower hospitalization costs than the SEN group. The authors conclude that preoperative administration of EIN significantly reduces the postoperative incidence of infectious complications in GC patients with cachexia, improves the inflammatory and immunological status, shortens hospital stay, and reduces healthcare costs [31]. 

Martínez González Á et al., in a real-world, observational retrospective cohort study, separated 134 patients with GC who underwent total gastrectomy into two groups: The first (79 patients) received a standard diet, while the second (55 patients) received a diet containing arginine, nucleotides, omega-3 fatty acids, and extra virgin olive oil (EIN group). The nutritional intervention was applied pre- and postoperatively for a mean period of 10 days. It was found that in the EIN group, hospital stay was significantly reduced. Compared to the control group, the number and duration of parenteral nutrition administration decreased by 21.1% and 33.2%, respectively. The risk of infectious complications was 70.1% lower in the EIN group compared to the control group. The number of postoperative complications, such as intestinal obstruction, suture detachment, blood transfusion, pleural effusion, acute renal failure, and surgical reoperation, was significantly lower at the level of 84.0, 90.9, 99.8, 90.9, 84.0, and 69.9%, respectively, compared with the control group. Finally, the group receiving EIN observed significantly less weight loss and a smaller decrease in postoperative albumin and serum cholesterol, demonstrating that EIN reduces postoperative complications and length of hospital stay while optimizing nutritional outcomes [32]. 

In the Li K et al. trial, it was found that early postoperative EIN improved immune function. This study aimed to evaluate the effect of EIN on immune function, inflammatory response, and nutritional status compared to standard enteral nutrition (SEN). A total of 124 patients with GC submitted to gastrectomy were randomized to receive early five days postoperative EIN (a formula enriched with arginine, glutamine, omega-3 fatty acids, and nucleotides) or SEN. The primary endpoints were CD4+ T-cells, CD3+ T-cells, CD4+/CD8+ counts, IgG, IgM, and IgA levels. Secondary endpoints included white blood cells, CRP, procalcitonin, tumor necrosis factor-α, IL-6 levels, and nutritional markers such as serum albumin, proalbumin, and transferrin concentration. There was a significant difference in the primary endpoints between the EIN and SEN groups. The percentage of CD4+ T-cells, CD3+ T-cells, and the numbers of CD4+/CD8+, IgG, IgM, and IgA were ultimately higher in the EIN group. Meanwhile, the WBC, CRP, and TNF-α level were finally significantly lower in the EIN group. However, there were no other significant differences in nutritional markers between the two groups. The study showed that early postoperative EIN significantly improves immune function and inflammatory response in gastric cancer patients undergoing gastrectomy [33]. 

Scislo L et al. studied the effect of postoperative administration of EIN (Reconvan, Fresenius Kabi, Bad Homburg, Germany) in 44 GC patients on the rate of postoperative complications and the 6-month and 1-year survival rates of patients. The results were compared with 54 patients who received SEN (Peptisorb, Nutricia, Schipol, The Netherlands). They found that the two groups did not differ in overall postoperative morbidity. Still, the rate of pulmonary complications and 60-day mortality were significantly lower in the group receiving EIN compared to the group receiving SEN. No difference in 6-month and 1-year survival was observed. The findings suggest that postoperative administration of EIN, although reducing respiratory complications and immediate postoperative mortality, does not improve survival compared with SEN. The reason may be related to the fact that the magnitude of this beneficial effect of EIN is too small to become statistically significant for the number of patients included in the study [34]. 

Bearing in mind that nutritional support with eicosapentaenoic acid could improve immune function and reduce catabolism in patients with advanced cancer, Ida et al., in a randomized phase III clinical trial, studied the effect of a diet rich in eicosapentaenoic acid in GC patients undergoing total gastrectomy. Patients were divided into a group receiving a SEN (61 patients) and a group receiving a SEN supplemented with eicosapentaenoic acid (ProSure^®^, Abbott GmbH Max-Planck-Ring 2 65205 Wiesbaden, Germany) (63 patients). The caloric content was 600 kcal, and the eicosapentaenoic acid was 2.2 g. The administration period was 7 days before and 21 days after surgery. It was found that the surgical morbidity in the two groups was 13% and 14%, respectively. No differences were found in weight loss one and three months after surgery (8.7 vs. 8.5%, respectively), suggesting that EIN enriched with eicosapentaenoic acid does not reduce weight loss after total gastrectomy for GC compared with SEN [35]. 

Klek S et al. studied the effect of EIN (formula enriched with arginine, glutamine, and omega-3 fatty acids) on the survival of GC patients. For this purpose, 99 patients operated for GC were randomized into two groups: 54 received SEN and 45 EIN. There was no difference in overall survival between the two groups; however, at the end of the third month, there were nine deaths in the SEN group and none in the EIN group. Univariate analysis showed that the EIN group had a lower risk of death, especially during the first year postoperatively. However, it did not affect the risk of death when the patients were analyzed overall. Moreover, a significant reduction in the risk of death was observed in the first six months postoperatively in the EIN group in stage IV patients, which supports the assumption that EIN does not significantly affect long-term survival. Nevertheless, this result suggests the need for further studies in the group of patients with stage IV GC [36].

Marano L et al. evaluated the effect of early postoperative EIN in 109 GC patients undergoing total gastrectomy. Patients were randomized to receive early postoperative EIN (formula supplemented with arginine, ω-3PUFAs omega-3 fatty acids, and ribonucleic acid) or to a control group receiving an isocaloric-isonitrogenous diet (SEN). The incidence of infectious complications and anastomotic leakage rate was found to be significantly lower in the EIN group. However, no significant differences in mortality rate w ere noticed, although the EIN group had reduced the length of hospital stay. Regarding the effect on immunological parameters, it was found that the postoperative CD4(+) T-cell count decreased in both groups, with the decrease being higher in the EIN group. The overall results suggest that early postoperative EIN improves clinical outcomes and some immunological parameters in patients undergoing gastrectomy for GC [37]. 

Fujitani K et al. evaluated the effect of preoperative EIN in 244 well-nourished GC patients. Patients were randomized into two groups of 117 and 127 patients, respectively, who received either EIN or a SEN for five days before surgery. The incidence of surgical site infection, infectious complications, and the overall postoperative morbidity rate did not differ significantly between the two groups, suggesting that five-day preoperative EIN is not advantageous regarding early clinical outcomes in well-nourished GC patients undergoing total gastrectomy [38]. 

Okamoto Y et al. evaluated the effect of preoperative EIN administration on cellular immunity, duration of systemic inflammatory response syndrome, and postoperative complications in 60 GC patients divided into two groups, the first of which received EIN with arginine and omega-3 fatty acids (30 patients). The second group received SEN (30 patients) for seven days preoperatively. They found that the incidence of postoperative infectious complications in the EIN group (6%) was significantly lower than that of the control group (28%) and that the duration of systemic inflammatory response syndrome in the EIN group was considerably shorter than that of the control group. They also noticed that although postoperative lymphocyte and CD4(+)T-cell counts were significantly decreased in both groups, the CD4(+)T-cell count on preoperative days 1 and 7 was significantly higher in the EIN group compared with the control group. This suggests that preoperative EIN enhances the immune status of patients, reduces the duration of systemic inflammatory response syndrome, and reduces the incidence of postoperative infectious complications [29]. 

Finally, Farreras N et al. randomized sixty patients with GC into two groups. The first group received early postoperative EIN with arginine, ω-3PUFAs omega-3 fatty acids, and ribonucleic acid (30 patients), while the second one received an isocaloric-isonitrogenous diet (30 patients). They found that patients in the IEN group had higher local hydroxyproline levels and significantly fewer episodes of surgical wound healing complications compared with controls, suggesting that early IEN in GC patients undergoing total gastrectomy increased hydroxyproline synthesis and improved surgical wound healing [27]. 

### 3.3. Metaanalyses

Several meta-analyses have been published on the efficacy of perioperative EIN administration in patients with GC undergoing gastrectomy (Table 2). However, it is paradoxical that the number of meta-analyses reporting on the effect of EIN compared with SEN is greater than the total number of clinical trials published. The conclusions of almost all these meta-analyses support the view that perioperative EIN administration is accompanied by a reduction in infectious complications and length of hospital stay compared with SEN but without a favorable effect on mortality. It appears that preoperative administration of EIN significantly improves the postoperative course as a result of the improvement of several metabolic parameters, as sufficient levels of immune nutrients are concentrated in plasma, resulting in the mitigation of inflammatory responses, an increase in cellular immune responses, and improvement of intestinal micro-diffusion and oxygenation. Preoperative EIN appears to achieve less dramatic results in malnourished patients while being effective in non-malnourished patients. However, the favorable effects of EIN in terms of many clinical parameters (hospital stay, surgical wound healing, the incidence of postoperative infections, etc.) need to be fully documented by large multicenter studies. Moreover, with an increase in the number of patients, the increase in the time of administration of EIN, as well as with better monitoring, there is the possibility of finding an improvement in patient survival. A description of the results of these meta-analyses is provided below. 

It has long been known that chronic inflammation is associated with the pathogenesis of neoplastic diseases and that it results in a worsening prognosis of cancer patients. On the other hand, the administration of ω-3 polyunsaturated fatty acids (ω-3PUFAs) (PUFAs) is recommended as an adjuvant anticancer therapy due to their anti-inflammatory properties. In this context, Mocellin MC et al. analyzed the findings of 9 randomized clinical trials involving 698 patients on the effect of ω-3PUFAs n-3 PUFAs contained in fish oil or added to a specific immunonutrition formula on the levels of serum inflammation markers in patients with GC. Of the nine studies included in the analysis, eight were conducted in surgical patients and one in patients receiving chemotherapy. In addition, fish oil was used as the sole intervention in four studies, and an immunonutrient formulation was administered in five studies. The results showed higher concentrations of albumin and proalbumin and lower concentrations of IL-6 and TNF-α in patients in the intervention group compared to the control group. The levels of total albumin, transferrin, and CRP did not improve. The results suggest that the administration of ω-3PUFAs n-3 PUFAs (fish oil) or ω-3PUFAs n-3 PUFAs added to an immune nutritional formulation has a favorable effect on serum inflammation markers in patients with GC undergoing surgery [17]. 

In their meta-analysis, Cheng Y et al. attempted to answer the question of whether EIN affects the biochemical and immunological parameters as well as the clinical outcome of patients operated on for GC. The meta-analysis ultimately included seven studies with 583 patients. The results showed that EIN whose postoperative administration exceeded the 7-day limit increased the level of CD4+, CD4+/CD8+, IgM, IgG, lymphocytes, and proalbumin levels. These positive effects were not visible in the first seven postoperative days. The levels of CD8+ and other serum proteins were not substantially improved. Some clinical parameters, such as systemic inflammatory response syndrome and postoperative complications, were significantly reduced in the EIN group. Lung infection rates, as well as hospital stay, did not improve regardless of the postoperative time. In this meta-analysis, EIN is found to improve cellular immunity, modify inflammatory responses, and reduce the rate of postoperative complications in GC patients undergoing surgery [21]. 

The meta-analysis by Song GM et al. included nine studies with 785 patients. The results showed that EIN increased IgA, IgG, IgM, CD4, CD3, CD4/CD8 levels and the number of natural killer cells. EIN decreased IL-6 and TNF-α level but did not alter total serum protein and CD8 count. It also did not reduce postoperative complications or hospitalization stay. According to the results of this meta-analysis, EIN enhances immune defenses and improves inflammatory responses in GC patients undergoing gastrectomy without improving clinical outcomes. Various factors such as patient heterogeneity, different timing of administration of EIN, small number of patients included, and other factors apparently reduce the power of the study [39].

In their study, Qiang H et al. included six clinical randomized trials with 308 GC patients who underwent early postoperative EIN and 298 patients who underwent SEN. On the seventh postoperative day, CD4, CD8, and CD4/CD8 ratios were significantly improved compared to SEN. Furthermore, postoperative complications, postoperative weight loss, and length of hospital stay were significantly lower in the early EIN group than in the early SEN group [40]. 

In their network meta-analysis, Song GM et al. compared the efficacy of different EIN formulas in GC patients undergoing gastrectomy. Eleven RCTs enrolling 840 patients were included. Pair-wise meta-analysis indicated that Arg+RNA+ω-3-FAs and Arg+Gln+ ω-3PUFAs ω-3-FAs are optimal for reducing infectious complications and length of hospital stay [41]. 

In their study, Niu JW et al. investigated the effect of EIN by performing a meta-analysis of 16 relevant clinical trials. They found that the risk of surgical site infection and hospital stay were significantly lower in the EIN group compared to the SEN group. Perioperative immunonutrition also significantly reduced white blood cell count and CRP levels. However, CD4+ T lymphocyte count and inflammatory cytokine levels were not significantly affected [42]. 

Matsui R et al. in their recent meta-analysis included 23 studies published up to 2022. This meta-analysis showed that EIN reduced infectious postoperative complications to a significant extent compared to SEN by approximately 30%. The results confirmed that nutritional perioperative EIN significantly reduces infectious complications in patients operated on for upper gastrointestinal tract cancer [43]. 

The meta-analysis by Adiamab et al. evaluated 16 studies that included 1387 patients (715 EIN group, 672 SEN—isothermal isonitrogenous food or normal diet), all undergoing gastrointestinal cancer surgery. Results showed that preoperative EIN reduced infectious complications and length of hospital stay compared to a control. However, it did not affect non-infectious complications or mortality [44].

In their meta-analysis, Shen J et al. included 35 clinical trials with 3692 patients who underwent surgery for digestive cancer (oesophageal, gastric, colon, and colorectal). The study revealed that the EIN group had a significantly reduced incidence of overall complications, particularly in infectious complications such as surgical site infection, occurrence of abdominal abscess, anastomotic escape, bacteremia, and duration of systemic inflammatory response syndrome. The duration of antibiotic administration was also shorter in the EIN group. Importantly, the study found no statistically significant effect of EIN on non-infectious complications and mortality. These findings strongly suggest that EIN is a safe and effective strategy for reducing overall complications, infectious complications, and hospital stays in patients undergoing surgery for digestive cancer [45]. 

In Li H et al.’s study, 12 randomized controlled clinical trials with a total of 10,422 GC patients were included. These findings strongly suggest using EIN to enhance the immunity of GC patients undergoing gastrectomy [46]. 

Finally, in the most recent meta-analysis by Li J et al., the authors evaluated 12 randomized and high-quality studies with 1124 patients (565 in the EIN group and 559 in the SEN group). They found that CD4+ levels, lymphocyte counts, transferrin concentrations, and systemic inflammatory response syndrome did not differ significantly between the two groups. However, CD8+, immunoglobulin G and M, and proalbumin concentrations, and CD4+/CD8+ ratio were significantly higher in the EIN group compared with the SEN group. In this meta-analysis, it was once again shown that EIN improves the immune function of patients operated on for GC [47]. 

**Table 2 biomedicines-12-02876-t002:** Μeta-analyses evaluating the role of perioperative EIN in GC patients.

Reference	Design	Results	Conclusion
Mocellin MC et al. 2018 [17]	9 trials, 698 patients. 4 trials used only fish oil as intervention and 5 trials used an immunonutrition formula.	Higher albumin and prealbumin concentrations and lower levels of IL-6 and TNF-α were noticed in the intervention group as compared to controls.	n-3 PUFAs supplementation has favorable effects on some inflammatory markers in patients operated-on for GC.
Cheng Y et al. 2018 [21]	Seven studies involving 583 patients.	Patients in whom postoperative administration exceeded the 7-day increased the level of CD4+, CD4+/CD8+, IgM, IgG, lymphocyte and proalbumin levels. SIRS and postoperative complications were reduced in the IEN group.	IEN improves cellular immunity, modifies inflammatory responses and reduces the rate of postoperative complications in GC patients undergoing surgery.
Song GM et al. 2015 [39]	Nine studies with 785 patients.	IEN increased IgA, IgG, IgM, CD4, CD3, CD4/CD8 levels as well as the number of NK cells. IEN decreased IL-6 and TNF-α level. IEN did not reduce postoperative complications or hospitalization stay.	EIN enhances immune defenses and improves inflammatory responses in GC patients undergoing gastrectomy without improving clinical outcomes.
Qiang H et al. 2017 [40]	Six studies 606 GC pts 308 early postoperative IEN and 298 on SEN.	On the 7th postoperative day, CD4, CD8, and CD4/CD8 ratios improved compared to SEN. Postoperative complications, weight loss, and length of hospital stay were lower in the early IEN group than in the early SEN group.	EIN enhances immune defenses and improves inflammatory responses in GC patients undergoing gastrectomy.
Adiamab A et al. 2019 [44]	16 studies, 1387 pts. 715 IMN & 672 normal diet (control group).	Preoperative IMN reduced infectious complications (*p* < 0.0001) and length of hospital stay. No effect on noninfectious complications or mortality.	Given the significant impact on infectious complications and a shorten length of hospital stay, preoperative IMN should be applied in routine practice in GI malignancies.
Song GM et al. [41]	Eleven RCTs enrolling 840 patients.	EIN Arg+RNA+ ω-3PUFAs ω-3-FAs, Arg+Gln+ ω-3PUFAs ω-3-FAs reduced infectious and noninfectious complications, and length of hospital stay.	Arg+RNA+ω-3-FAs and Arg+Gln+ω-3-FAs are optimal for reducing infectious complications and length of hospital stay.
Matsui R et al. 2024 [43]	23 studies published up to 2022.	IEN reduced infectious postoperative complications to a statistically significant extent compared to SEN by approximately 30%.	Nutritional perioperative IEN reduces infectious complications in patients operated on for upper GI cancer.
Shen J et al. 2022 [45]	35 clinical trials with 3692 patients.	IEN group infectious complications and duration of systemic inflammatory response syndrome) were lower compared to the control group.	IEN is safe and effective in reducing overall complications, infectious complications, and hospital stays.
Li H et al. 2023 [46]	Twelve randomized controlled clinical trials with a total of 10,422 GC pts.	Odds ratio value: 0.23 (95% CI: 0.09–0.59).	The findings strongly suggest the use of EIN to enhance the immunity of GC pts undergoing gastrectomy.
Li J et al. 2024 [47]	Twelve randomized clinical trials with 1124 pts (EIN:565, SEN: 559).	CD8+, IgG, IgM, proalbumin, and CD4+/CD8+, were higher in the EIN group than in the SEN group.	IEN is effective in improving immune function in patients who have undergone GC surgery.

IL-6 = Interleukin-6, TNF-α = Tumor Necrosis Factor-α, NK cells = Natural Killer cells, SIRS = Systemic inflammatory response syndrome. ω-3PUFAs = ω-3 Fatty Acids.

## 4. Discussion

It is well known that a significant proportion of patients with GC present a clinical picture of malnutrition, which is expected to worsen following gastrectomy for the underlying malignancy. This causes a decrease in immune responses, an increase in inflammatory responses, and a decrease in chemical and cellular immunity in the immediate postoperative period [48]. 

EIN is a method used in clinical application in recent years to improve the immune function of patients. It refers to adding glutamine, arginine, ω-3PUFAs ω-3 unsaturated fatty acids, nucleotides, and other immunomodulatory nutrients to the standard enteral nutrition formula. These immune modulatory components can stimulate the immune cells of the body and improve the immunity of the body after being absorbed by the body. The content of EIN HIN varies considerably between formulations, but some common features include the enrichment in arginine and/or glutamine amino acids, the nucleotides, and the different amounts of ω-3PUFAs. omega-3 fatty acids. 

This review aims to clarify the effects of EIN on the clinical outcome of GC patients undergoing gastrectomy and whether its implementation improves outcomes. From the review of clinical trial data and relevant meta-analyses, it can be safely concluded that by applying perioperative EIN we can reduce the inflammatory process while providing significant clinical benefits by reducing postoperative infectious complications and the length of hospital stay. However, it should be stressed that patients included in these trials displayed huge differences in their responses to different combinations of immunonutrient supplementation. This should be kept in mind when trying to transfer this information into clinical practice by selecting the most suitable regime. In the clinical studies and meta-analyses reported, the authors also used different clinical and laboratory response indices to demonstrate the effectiveness of EIN on patients’ clinical outcomes. Adequate justification is not provided for why a particular study used a different methodology rather than another. However, these differences added other elements that may enhance the scientific value of the studies’ results, as the clinical and laboratory outcomes are more widely considered.

According to the results of the clinical studies and meta-analyses cited, perioperative application of EIN primarily preoperatively enhances the function of the intestinal mucosal barrier, mitigates systemic inflammatory responses, enhances cellular immunity function, and significantly reduces postoperative complications, instilling confidence in its potential to improve patient outcomes [49]. It is of interest that despite the favorable results obtained and described in all available trials, no prolongation of patient survival was found apparently probably because mortality is influenced by many other factors such as muscle mass, age, coexisting conditions, and a positive history of smoking and ethyl alcohol consumption, conditions that are not altered by taking EIN. This should be an area of intensive future research, with large numbers of patients included in multicenter clinical trials so that any borderline differences become statistically significant.

Most clinical guidelines recommend EIN in oncology patients at risk of malnutrition or in patients with malignant digestive tumors who are about to undergo surgical removal of the cancer [50]. For example, the European Society for Parenteral and Enteral Nutrition (ESPEN) recommends that patients undergoing surgery for upper digestive cancer should receive an EIN to reduce post-operative infectious complications. 

Among the most important factors predicting patients’ clinical responses to perioperative immunonutrition seems to be their nutritional status. In patients with satisfactory nutritional status, immunonutrition does not seem to significantly affect the clinical outcome. On the contrary, negatively affected nutritional status is a factor that positively influences the results of EIN. Providing EIN orally or by tube for 5 to 7 days is recommended preoperatively in all malnourished and non-malnourished patients who are planning to undergo surgery for GC. In addition, it should be continued postoperatively in patients who were preoperatively malnourished, and for 5 to 7 days if there are no complications or until oral food intake is restored to at least 60% of total nutritional requirements. A post-operative intake of 1000 kcal/day in addition to the regular diet is recommended. EIN is contraindicated in patients with sepsis and concomitant hemodynamic disturbances. 

Preoperatively, the oral approach should be preferred. In contrast, the enteral tube approach should be used if oral administration is impossible or if the patient has lost more than 10% of their average weight [51]. Post-operatively, oral intake is preferred if surgery allows, and the upper gastrointestinal tract is functional. If not, as in the immediate postoperative period after a gastrectomy for cancer, the enteral tube approach should be preferred. An example of an EIN is the Nestle Oral Impact^®^ formulation (237 or 500 mL cans) for feeding via catheter, which contains 18 or 28 g of protein and yields 334 or 505 kcal.

## 5. Conclusions and Future Directions

From what has been reported, it is concluded that perioperative use of EIN in patients operated-on for GC reduces inflammatory cytokine levels, postoperative infectious complications, and the length of hospital stay. Although it does not seem to increase the survival odds, this possibility should be studied more intensively. An important observation is whether one or a combination of more immunonutrients achieves better results. Although the individual authors do not state this, the authors of this review believe that the combination of the nutritional elements detailed in the text may provide the best results and, therefore, be the recommended one in daily clinical practice. EIN should be continued postoperatively in preoperatively malnourished patients for 5 to 7 days if there are no complications or as long as required until oral feeding is restored, providing at least 60% of nutritional needs. Combining EIN with physical activity is recommended as it is most effective in increasing muscle blood flow and protein assimilation while reducing inflammatory phenomena [50]. Most authors agree that the most suitable time for EIN administration should be as early as possible. Furthermore, no significant benefit is expected from its administration in preoperative patients with a satisfactory level of nutrition or in patients who are expected to be normally fed five days after surgery. Furthermore, no benefit is expected in patients in whom enteral feeding is contraindicated, nor in patients who cannot be helped with any form of nutritional support for their underlying disease.

An important element of the whole effort is the need for a collaborative approach between the oncology department and the nutrition unit in properly assessing the nutritional status of GC patients. This can be achieved by taking a rapid nutritional history, determining anthropometric parameters, and a basic analytical determination using one of the existing assessment tools [49]. Early demonstration of the presence of malnutrition is particularly important for patient outcomes. 

Finally, the importance of patient selection for EIN administration cannot be overstated. Given the relatively high cost of EIN, it is crucial to identify the patients who would benefit the most from its use. Furthermore, because it is possible that other substances may also have immunomodulatory effects, research should be directed toward this area. When considering the economic impact, the reduction in hospitalization costs (reduction in hospital days) and the reduction in complications (reduction in costs of antibiotic maintenance, etc.) should be taken into account.

## Figures and Tables

**Figure 1 biomedicines-12-02876-f001:**
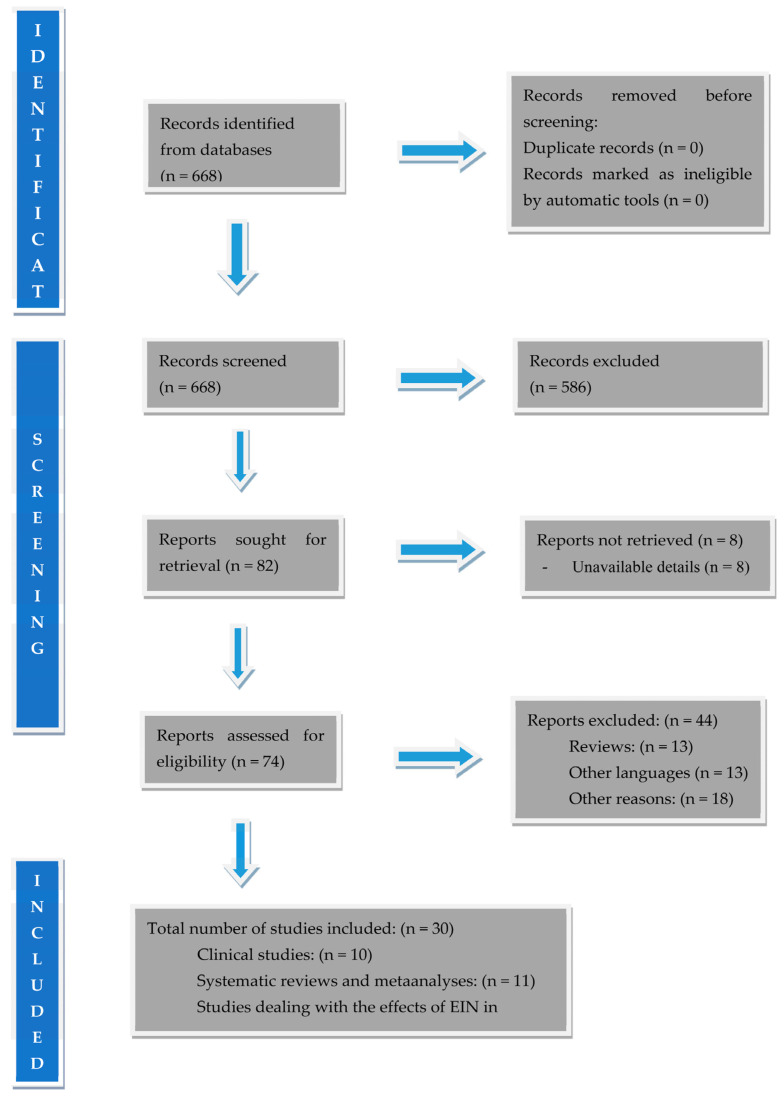
PRISMA flow diagram used in this systematic review.

**Table 1 biomedicines-12-02876-t001:** Clinical trials on EIN in GC patients.

Author/Year	Type of Study	Aim	Groups/Numbers	Results	Conclusion
Yu J et al. 2024 [31]	Prospective Randomized Controlled Trial	Evaluation of the incidence of infectious complications, immune status, hospital stay, and healthcare cost.	One hundred twelve patients with GC and cachexia received either preoperative EIN (n = 56) or standard enteral nutrition support (SEN, n = 56).	IN vs. SEN group: Significantly lower incidence of postoperative and overall infectious complications, lower levels of WBC), CRP, IL-6, and higher levels of lymphocytes and IgA). Clinical outcome: Less weight loss, shorter duration of antibiotic use, hospital stay, and total hospital costs compared to the SEN group.	Preoperative ΙΝ in patients with GC and cachexia reduces the incidence of infectious complications, improves immune status, shortens hospital stay, and reduces healthcare costs.
Martínez González Á et al. 2024 [32]	Observational retrospective cohort study 134 patients	To assess pre-and postoperatively the effectiveness of EIN compared to SEN in patients undergoing GC surgery.	134 patients Group A: (n = 79) SEN Group B (n = 55) EIN with arginine, nucleotides, ω-3PUFAs omega-3 fatty acids, and extra virgin olive oil.	Hospital stay: IEN group: 34% reduction Patients needed parenteral nutrition: EIN group: 21.1% reduction Risk of infectious complications: 70.1% less in the EIN group. Also, less weight loss, blood transfusions, and surgical re-intervention.	IEN reduces postoperative complications and hospital stays and optimizes nutritional outcomes.
Li K. et al. 2020 [33]	Prospective Randomized Controlled Trial	Evaluation of postoperative immune status.	124 GC pts. Early 5ds postoperative EIN vs. SEN	Higher CD4+ T-cells CD3+ T-cells and CD4+/CD8+ IgG, IgA, IgM in EIN group. Lower WBC, CRP, PCT, and TNF-α.	Early postoperative EIN improves immune function and inflammatory responses in GC pts undergoing gastrectomy.
Scislo L et al. 2018 [34]	Randomized clinical Trial	Postoperative Immunonutrition vs. SEN.	98 pts EIN group (n = 44) vs. SEN (Peptisorb) (n = 54)	Postoperative morbidity: No difference Rate of pulmonary complications: Lower in the EIN group. Sixty-day mortality: lower in the EIN group 6th and 12th-month survival: no difference.	Postoperative IEN reduces respiratory complications and postoperative mortality in comparison to standard EN. No improvement in the 6-mo and 1-yr survival.
Ida S et al. 2017 [35]	Randomized phase III clinical trial	Eicosapentaenoic acid-rich nutrition vs. standard diet.	124 patients: Safety: 61 SEN vs. 63 EIN Efficacy: 60 SEN vs. 63 EIN. Seven days before and 21 days after surgery.	Surgical morbidity rate: No difference between the groups (13% vs. 14%) Median bodyweight loss at one month and three months after gastrectomy: No significant differences between the groups.	IEN based on an eicosapentaenoic acid-enriched oral diet did not reduce body weight loss after total gastrectomy compared with a standard diet.
Klek S et al. 2017 [36]	Randomized controlled clinical trial	EIN enriched with arginine, glutamine, and ω-3PUFAs vs. Standard diet	99 patients. 54 SEN vs. 45 EIN. Short- and long-term (5 y) survival was analyzed.	Overall survival: No differences. End of the third month: Nine deaths in the SEN vs. no deaths in the EIN. Univariate analyses: EIN group: lower risk during the first year. Also there was a reduction in the risk of death in the EIN group during the first six mos after surgery (stage IV GC). EIN did not influence the risk of dying when patients were analyzed together.	No beneficial effect of IEN on long-term survival. A positive impact on stage IV GC patients suggests the need for further studies.
Marano L et al. 2013 [37]	Randomized controlled clinical trial	Early postoperative EIN vs. an isocaloric -isonitrogenous diet.	109 patients. Early postoperative IEN (with arginine, ω-3PUFAs omega-3 fatty acids, and ribonucleic acid (54 pts) vs. An isocaloric -isonitrogenous diet (55 pts).	Incidence of postoperative infectious complications: Lower in EIN than in the control group. Anastomotic leak rate: Lower in the IEINgroup. Mortality rate: No differences. Length of hospitalization: Reduced in EIN group Cellular immunity: CD4+ T-cell decreased in both groups. Higher reduction in the EIN group.	EIN improves clinical and immunological outcomes in patients with GC undergoing gastrectomy.
Fujitani K et al. 2012 [38]	Prospective randomized clinical trial	Clinical effects of preoperative EIN in well-nourished GC patients undergoing total gastrectomy.	Total: 244 patients EIN group: 127 pts. Control group: 117 pts.	Surgical-site infections: No significant differences Infectious complications: No significant differences. Overall postoperative morbidity rate: No significant differences.	Five-day preoperative EIN failed to show benefits in terms of clinical outcomes in well nourished patients with GC undergoing total gastrectomy.
Okamoto Y et al. 2009 [29]	Randomized clinical trial	Evaluation of the effect of preoperative EIN on postoperative complications in patients with GC.	EIN group (n = 30) supplemented with arginine and ω-3PUFAs. Control group (n = 30). Standard formulas preoperatively seven days.	Postoperative infectious complications: Significantly lower in the EIN group.	Preoperative EIN supplemented with arginine and ω-3PUFAs decreased the postoperative infectious complications.
Farreras N et al. 2005 [27]	Randomized clinical trial	Effect of early postoperative EIN on wound healing in patients with GC undergoing gastrectomy.	EIN supplemented with arginine, ω-3PUFAs omega 3 fatty acids, and RNA: 30 patients Isocaloric-isonitrogenous control: 30 patients.	EIN: Higher local hydroxyproline levels, Lower episodes of surgical wound healing complications compared with the control group.	Early postoperative EIN with a formula increased hydroxyl-proline synthesis and improved surgical wound healing in GC.

SEN = Standard Enteral Nutrition, EIN = Enteral Immunonutrition, EN = Enteral Nutrition, GC = Gastric Cancer, WBC = White Blood Cells, CRP = C-reactive protein, IL-6 = Interleukin-6, IgA = Immunoglobulin A, ω-3PUFAs = ω-3 fatty acids.

## Data Availability

Not applicable.

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
