# Peer review of "Immunonutrition in Operated-on Gastric Cancer Patients: An Update"

_biomedicines, 2024, doi:10.3390/biomedicines12122876_

Round 1
Reviewer 1 Report
Comments and Suggestions for Authors
John et al. submitted the manuscript entitled: Immunonutrition in operated-on gastric cancer patients: An update, in which the authors summarized recent advances on immunonutrient supplementation for preoperative gastric cancer patients. The authors focused on recent trials and meta-analyses and concluded the efficacy of immunonutrient supplementation from the aspects of immune responses and clinical responses and provided detailed information of these references. Generally, this topic will be of interest to potential readers of Biomedicines, but I believe more detailed analysis should be included.
My comments are as follows.
1. Page 3, Figure 1 and Page 2, Methods: It is advised to include exclusion criteria in methods and figures. The authors can include reasons for each excluded and unretrieved records (unavailable details, language not in English and etc.)
2. Patients have huge differences when responded to different combinations of immunonutrient supplementation. It is advised to include the details in table 1 and 2. Besides, citations are missing in these tables.
3. Since this manuscript is an update. Are the new references all included in existing meta-analysis? If not, will the new references change the conclusions of these meta-analysis?
4. Discussion section: I believe these topics are worth discussing: 4a) In table 1, different immune response indicators for different indications were used in these meta-analyses. Why did the authors use these indications? Are these indicators reasonable? 4b) For different indications in operated-on gastric cancers, is there some type(s) of supplementation that appeared essential in most of the cases? 4c) Any factors were identified as essential for clinical responses to immunonutrient supplementation?
Author Response
John et al. submitted the manuscript entitled: Immunonutrition in operated-on gastric cancer patients: An update, in which the authors summarized recent advances on immunonutrient supplementation for preoperative gastric cancer patients. The authors focused on recent trials and meta-analyses and concluded the efficacy of immunonutrient supplementation from the aspects of immune responses and clinical responses and provided detailed information of these references. Generally, this topic will be of interest to potential readers of Biomedicines, but I believe more detailed analysis should be included.
The authors of the review would like to express their sincere thanks to the article's reviewers for their efforts and pertinent comments, which improved the article considerably.
- Page 3, Figure 1 and Page 2, Methods: It is advised to include exclusion criteria in methods and figures. The authors can include reasons for each excluded and un-retrieved records (unavailable details, language not in English and etc.)
Answer: Some exclusion criteria were added. Reasons for un-retrieved records were also included in figure 1.
- Patients have huge differences when responded to different combinations of immunonutrient supplementation. It is advised to include the details in table 1 and 2. Besides, citations are missing in these tables.
Answer: This remark was added in the discussion part along with possible explanations and suggestions.
Citation numbers were added in tables 1 and 2.
- Since this manuscript is an update. Are the new references all included in existing meta-analysis? If not, will the new references change the conclusions of these meta-analyses?
Answer: To the best of our knowledge, all the available relevant meta-analyses were included in the article.
- Discussion section: I believe these topics are worth discussing:
4a) In table 1, different immune response indicators for different indications were used in these meta-analyses. Why did the authors use these indications? Are these indicators reasonable?
Answer: This critical observation was included in the discussion section.
4b) For different indications in operated-on gastric cancers, is there some type(s) of supplementation that appeared essential in most of the cases?
Answer: This important observation is also commented on in the conclusion section.
4c) Any factors were identified as essential for clinical responses to immunonutrient supplementation?
Answer: The most important clinical factors are listed in the discussion section.
REVIEWER 2.
The paper “Immunonutrition in operated-on gastric cancer patients: An update” describes and analyses the clinical results and the findings of relevant meta-analyses of the application of enteral immune nutrition in gastric cancer patients, emphasize the importance of this therapeutic intervention for disease progression, and attempts to provide practical guidelines for applying enteral immune nutrition in daily clinical practice. Below are my relevant comments.
The authors of the review would like to express their sincere thanks to the article's reviewers for their efforts and pertinent comments, which improved the article considerably.
Comments:
Q1. Line 16-18. The statement seems to be ambiguous, enhance the rate of postoperative complications? Enhance the duration of hospitalization?
Answer: EIN reduces the rate of postoperative complications and the duration of hospitalization. This was clarified in the abstract section.
Q2. The term enteral immune nutrition refers to the perioperative administration of nutritional preparations containing, among others, specific ingredients such as glutamine, omega-3 polyunsaturated fatty acids, and arginine. Where does this recipe come from?
Answer: The ESPEN guidelines on nutrition in cancer patients refer to arginine, omega-3 fatty acids and nucleotides under the heading “immunonutrition”. This reference was added to the list of references. Arends J, Bachmann P, Baracos V, Barthelemy N, Bertz H, Bozzetti F, Fearon K, Hütterer E, Isenring E, Kaasa S, Krznaric Z, Laird B, Larsson M, Laviano A, Mühlebach S, Muscaritoli M, Oldervoll L, Ravasco P, Solheim T, Strasser F, de van der Schueren M, Preiser JC. ESPEN guidelines on nutrition in cancer patients. Clin Nutr. 2017;36(1):11-48. doi: 10.1016/j.clnu.2016.07.015. PMID: 27637832.
Q3. Numerous studies have demonstrated that basic amino acids, such as arginine, exhibit immunoactive properties. What is the underlying mechanism of action? Why does lysine, also a basic amino acid, display comparatively reduced immunological activity?
Answer: The immunological effects of arginine are reported in the introduction section. The neoplasm-enhancing actions of lysine with the relevant references were also added in this section.
Q4. The immunomodulatory mechanisms of glutamine, omega-3 polyunsaturated fatty acids, and arginine (line 440~500) should be move to the introduction section.
Answer: These paragraphs were moved to the introduction section.
Q5. Is there a contradiction between the repeated emphasis in this article that immunonutrition supplements can improve immune activity and reduce complications, and the lack of effect on patient survivors?
Answer: Of course, there is. The discussion section discusses this contradiction and suggests ways of further clarification.
Q6. n-3 PUFAs or ω-3 PUFAs? The statements in the text should be uniform.
Answer: Actually, it is the same. In the text, we used the term ω-3PUFAs
Q7. There should be no references in the conclusion section.
Answer: Reference 56 was deleted. Reference 57 was transferred to the discussion section.

Reviewer 2 Report
Comments and Suggestions for Authors
The paper “Immunonutrition in operated-on gastric cancer patients: An update” w describes and analyses the clinical results and the findings of relevant meta-analyses of the application of enteral immune nutrition in gastric cancer patients, emphasize the importance of this therapeutic intervention for disease progression, and attempts to provide practical guidelines for applying enteral immune nutrition in daily clinical practice. Below are my relevant comments.
Comments:
Q1. Line 16-18, The statement seems to be ambiguous, enhance the rate of postoperative complications? Enhance the duration of hospitalization?
Q2. The term enteral immune nutrition refers to the perioperative administration of nutritional preparations containing, among others, specific ingredients such as glutamine, omega-3 polyunsaturated fatty acids, and arginine. Where does this recipe come from?
Q3. Numerous studies have demonstrated that basic amino acids, such as arginine, exhibit immunoactive properties. What is the underlying mechanism of action? Why does lysine, also a basic amino acid, display comparatively reduced immunological activity?
Q4. The immunomodulatory mechanisms of glutamine, omega-3 polyunsaturated fatty acids, and arginine (line 440~500) should be move to the introduction section.
Q5. Is there a contradiction between the repeated emphasis in this article that immunonutrition supplements can improve immune activity and reduce complications, and the lack of effect on patient survivors?
Q6. n-3 PUFAs or ω-3 PUFAs? The statements in the text should be uniform.
Q7. There should be no references in the conclusion section.
Author Response
REVIEWER 2.
The paper “Immunonutrition in operated-on gastric cancer patients: An update” describes and analyses the clinical results and the findings of relevant meta-analyses of the application of enteral immune nutrition in gastric cancer patients, emphasize the importance of this therapeutic intervention for disease progression, and attempts to provide practical guidelines for applying enteral immune nutrition in daily clinical practice. Below are my relevant comments.
The authors of the review would like to express their sincere thanks to the article's reviewers for their efforts and pertinent comments, which improved the article considerably.
Comments:
Q1. Line 16-18. The statement seems to be ambiguous, enhance the rate of postoperative complications? Enhance the duration of hospitalization?
Answer: EIN reduces the rate of postoperative complications and the duration of hospitalization. This was clarified in the abstract section.
Q2. The term enteral immune nutrition refers to the perioperative administration of nutritional preparations containing, among others, specific ingredients such as glutamine, omega-3 polyunsaturated fatty acids, and arginine. Where does this recipe come from?
Answer: The ESPEN guidelines on nutrition in cancer patients refer to arginine, omega-3 fatty acids and nucleotides under the heading “immunonutrition”. This reference was added to the list of references. Arends J, Bachmann P, Baracos V, Barthelemy N, Bertz H, Bozzetti F, Fearon K, Hütterer E, Isenring E, Kaasa S, Krznaric Z, Laird B, Larsson M, Laviano A, Mühlebach S, Muscaritoli M, Oldervoll L, Ravasco P, Solheim T, Strasser F, de van der Schueren M, Preiser JC. ESPEN guidelines on nutrition in cancer patients. Clin Nutr. 2017;36(1):11-48. doi: 10.1016/j.clnu.2016.07.015. PMID: 27637832.
Q3. Numerous studies have demonstrated that basic amino acids, such as arginine, exhibit immunoactive properties. What is the underlying mechanism of action? Why does lysine, also a basic amino acid, display comparatively reduced immunological activity?
Answer: The immunological effects of arginine are reported in the introduction section. The neoplasm-enhancing actions of lysine with the relevant references were also added in this section.
Q4. The immunomodulatory mechanisms of glutamine, omega-3 polyunsaturated fatty acids, and arginine (line 440~500) should be move to the introduction section.
Answer: These paragraphs were moved to the introduction section.
Q5. Is there a contradiction between the repeated emphasis in this article that immunonutrition supplements can improve immune activity and reduce complications, and the lack of effect on patient survivors?
Answer: Of course, there is. The discussion section discusses this contradiction and suggests ways of further clarification.
Q6. n-3 PUFAs or ω-3 PUFAs? The statements in the text should be uniform.
Answer: Actually is the same. In the text we used the term ω-3PUFAs
Q7. There should be no references in the conclusion section.
Answer: Reference 56 was deleted. Reference 57 was transferred to the discussion section.